# Prevalence of childhood polyvictimization, mental health outcomes, and associated risk and protective factors in Ethiopia and Uganda refugee settings

Stella Muthuri[1]*, George Odwe[1], Francis Obare[1], Peter Kisaakye[1], Dagim Habteyesus[2], Gloria Seruwagi[1], Yohannes Wado[3], Yadeta Bacha[3,4], Bonnie Wandera[3], Caroline Kabiru[3], Chi-Chi Undie[1]

1 Population Council, Nairobi, Kenya, 2 Population Council, Addis Ababa, Ethiopia, 3 African Population and Health Research Center (APHRC), Nairobi, Kenya, 4 Haramaya University, Harar, Ethiopia

* smuthuri@popcouncil.org

## Abstract

### Background

Children exposed to polyvictimization (exposure to multiple types of victimization), experience severe impacts on health and well-being across the life course. While the prevalence, risk factors, and health consequences of polyvictimization have been well documented in lower-middle-income countries, there remains a significant knowledge gap regarding the situation in humanitarian contexts, particularly in refugee settings.

### Objectives

We examined the prevalence, risk, and protective factors of polyvictimization and its association with mental health outcomes in refugee settings in Uganda and Ethiopia.

### Data and methods

We utilized data from the Uganda Humanitarian Violence Against Children and Youth Survey (HVACS) 2022, which included 1,338 females and 927 males, and the Ethiopia HVACS 2024, which comprised 1,937 females and 1,536 males. These were representative, cross-sectional household surveys of females and males aged 13–24 years living in refugee settlements in Uganda and Ethiopia. Analysis entailed cross-tabulation with chi-square test and estimation of a multivariate logistic regression model.

### Results

In both settings, males experienced higher rates of polyvictimization than females (49% vs.30% in Uganda and 33% vs. 29%. in Ethiopia). Several risk factors for

**Data availability statement:** The Uganda Humanitarian Violence Against Children and Youth Dataset is publicly available here: https://knowledgecommons.popcouncil.org/hubs_humanitarian/17/ The Ethiopia Humanitarian Violence Against Children and Youth Dataset is available here: https://knowledgecommons.popcouncil.org/hubs_humanitarian/42/.

**Funding:** This study was funded by the United Kingdom's Foreign, Commonwealth and Development Office (FCDO) through the Baobab Research Program Consortium (RPC), with the Population Council Inc as the primary awardee under contract number PO8612. The funder did not play any role in the design of the study, collection, analysis and interpretation of the data, writing of the manuscript, or the decision to submit the manuscript for publication.

**Competing interests:** The authors have declared that no competing interests exist.

childhood polyvictimization were identified at the individual level, including having a disability and endorsing intimate partner violence against women, and at the family level, such as having a difficult relationship with one's father and household food insecurity. Conversely, living in a female-headed household was found to be protective against childhood polyvictimization across both settings. Additionally, polyvictimization was significantly associated with higher odds of mental distress, self-harm, and suicidal ideation/attempts among females and males in both Uganda and Ethiopia refugee settings.

## Conclusion

Our findings confirm that childhood polyvictimization is prevalent among refugee populations in Uganda and Ethiopia, with nuanced insights into the risk and protective factors. The challenging circumstances within these refugee settings may exacerbate polyvictimization and its consequences. Therefore, it is essential to prioritize preventive measures alongside response strategies in addressing polyvictimization in these contexts, starting much earlier in the life course.

## Introduction

Violence against children (VAC) is a serious, pervasive, and complex public health and human rights issue that affects nearly 50% of children globally with an estimated one billion children reporting experiencing violence yearly [1,2]. Polyvictimization, defined as having experienced several different forms of violence over the course of a given period, is widespread and is distinctly different from exposure to a single, or repeated episode of the same form of victimization [3,4]. A systematic review of the evidence from low- and lower-middle-income countries (LMICs) found a prevalence of polyvictimization among children and adolescents ranging from 0.3% in El Salvado to 74.7% in Vietnam, with an overall estimate of 38.1% [5]. A meta-analysis examining the prevalence of family polyvictimization found that 10% of the general population and 36% of a clinical population experienced co-occurring family victimization [6].

Polyvictimization is associated with a broad range of individual-, family- and community-level factors. Whereas some studies report higher prevalence of polyvictimization in boys than girls [7,8], others show gender parity in the prevalence of polyvictimization. For instance, a study in Ethiopia revealed a comparable but high prevalence among adolescent girls and boys at 50% and 53%, respectively [9]. Analysis of VAC surveys in LMICs found that school enrollment was protective against lifetime polyvictimization for females in Haiti and males in Kenya, Haiti, and Cambodia, but a risk factor for females in Nigeria [10]. Additionally, not living with a biological father increased the risk of polyvictimization among females in Cambodia, Malawi, and Nigeria, while absence of a biological mother was a risk factor for females in Kenya. Other studies have highlighted age, disability, orphanhood, household food insecurity, and weak social networks as risk factors for polyvictimization [11–13].

Several studies have linked polyvictimization to severe health consequences and risky behaviours [14–16]. Analysis of VAC survey in El Salvador showed polyvictimization was associated with greater odds of mental distress, and suicide ideation [17]. In Ethiopia, polyvictimization was associated with internalizing problems such as depression, anxiety and reduced attachment to parents for both female and male adolescents, a relationship mediated by resilience (measure based on a12-item scale that examines four components of social ecological resilience including individual, relational, community, and cultural aspects) among girls only [9]. A meta-analysis of data from several LMICs has similarly shown the association between polyvictimization and an increased likelihood of mental health problems (e.g., symptoms of post-traumatic stress disorders [PTSD], depression, and anxiety) and engagement in health risk behaviours such as substance use [5]. Additionally, few studies, largely in high income countries, have explored the relationships between different typologies or severities of polyvictimization and risky behaviours. For instance, exposure to violent home environments and high levels of verbal/social peer victimization have been associated with increased odds of adolescent substance use [18]. While a considerable body of research on childhood polyvictimization exists, including in LMICs, little is known regarding the situation in humanitarian contexts in these countries, particularly in refugee settings. Children and adolescents in refugee settings often face unique experiences distinct from those in non-humanitarian contexts [19,20]. Unlike most disadvantaged children in development settings, including hosting communities, children living in refugee settings face layered and unique risks across the pre-conflict, conflict, and post-arrival phases. These include exposure to pre-migration trauma such as conflict and persecution; disruption of protective factors through separation from parents and families and loss of community support structures; and post-arrival stressors like overcrowded, insecure living environments with limited and overstretched support systems [21]. These realities underscore the need for targeted research and programming to prevent and identify polyvictimized individuals and link them to care. This is particularly true since children in refugee contexts are at a high risk of experiencing multiple forms of abuse. Our prior cross-sectional representative household surveys conducted in Uganda's refugee settlements revealed an overlap of different types of violence in childhood. About half (50%) of males and 43% of females aged 18–24 years had experienced VAC at least one form of either sexual, physical, or emotional violence prior to age 18, while 65% of males and 49% of females aged 13–17 years reported ever experiencing VAC in their lifetime [22]. Further, in Uganda, among respondents aged 18–24, a substantial proportion reported that their first experience of violence occurred after arrival in the refugee settlements: 67% of females and 43% of males for physical violence, 73% of females and 53% of males for sexual violence, and 89% of females and 43% of males for emotional violence. Peers accounted for the largest share of perpetrators of physical violence (25% among females and 46% among males), and current or former spouses, boyfriends, girlfriends, or romantic partners the largest share of sexual violence perpetrators (31% among females and 29% among males) [23]. In Ethiopia's refugee camps, 32% of females and 26% of males aged 18–24 years experienced VAC before age 18, while among those 13–17 years of age, 37% of females and 44.8% of males experienced VAC in their lifetime [24]. Further, 32.3% of females and 65.9% of males who experienced sexual violence before age 18 reported that the first incident happened after arrival in the refugee camps. The figures were 74.7% of females and 43.9% of males for the first incident of physical violence happening after arrival [24]. These findings point to heightened risks of experiencing multiple forms of violence in refugee settings. In both countries, unsurprisingly, experiencing any VAC was associated with mental distress, self-harm, and suicide ideation in these refugee settings, with some notable variations by sex and types of violence experienced [24,25].

We estimate the prevalence of and assess risk and protective factors for childhood polyvictimization, defined as exposure to multiple forms of VAC (sexual, physical, and/or emotional) and/or witnessing physical violence within the home or community. Guided by an ecological framework [26], we examined the associations between childhood polyvictimization and: a) individual factors (e.g., age, disability status, endorsement of intimate partner violence against women (IPVAW) and school enrollment); b) family factors (e.g., orphanhood, household food insecurity, difficult parental relationships, and living in a female-headed household); and c) community factors (e.g., supportive friendships or adults in the community, and participation in community groups) among 13–24-year-old females and males in Uganda and Ethiopia refugee

settings. We also examined the association between childhood polyvictimization and mental health outcomes (moderate-to-severe mental distress, self-harm, and suicide ideation/attempt), including whether these relationships differed by sex across refugee settings in Uganda and Ethiopia. Understanding the prevalence of childhood polyvictimization, its risk factors, and its association with mental health outcomes could inform the design of effective violence prevention and response strategies in refugee settings.

## Methods

### Data sources

We used data from the 2022 Uganda Humanitarian Violence Against Children and Youth Surveys (HVACS), and from the 2024 Ethiopia HVACS collected. These are representative, cross-sectional, household surveys in refugee settings in Uganda and Ethiopia. Detailed information on the Uganda HVACS study design and sampling procedures, has been published elsewhere [27]. The Uganda and Ethiopia HVACS utilized the standard Violence Against Children and Youth Survey (VACS) methodology [28], adapted for humanitarian settings following implementation guidance [29].

### Study population and sampling

Similar to the standard VACS, both Uganda and Ethiopian HVACS conducted interviews with female and male children and young people aged 13–24 years living in refugee settings (all 13 refugee settlements in Uganda and 20 out of 23 refugee camps in Ethiopia). Three refugee camps in Ethiopia were excluded either due to security concerns or logistical challenges. A three-stage cluster sampling process was used in both countries to randomly select zones, households, and one eligible participant from each household. A split sampling design (that is, assigning separate zones for female and male samples within each settlement or camp to protect participant confidentiality and reduce the risk of interviewing both a perpetrator and survivor in the same community) was used.

### Data collection

The Uganda HVACS was conducted in April-May 2022 whereas the Ethiopia HVACS was conducted between December 2023 and March 2024. In both countries, the HVACS comprised two questionnaires: household and participant. The household questionnaire assessed the socioeconomic conditions of the household and basic demographic information and was administered to the head of household. The participant questionnaire, which included male and female versions, was administered to one eligible 13–24-year-old participant per household. The participant questionnaire was designed to measure experiences of VAC (emotional, physical, and sexual), including witnessing physical violence at home or in the community, risk or protective factors as well as the consequences of such violence. The study tools were translated from English into the local most used languages. Data collection was done electronically using the Open Data Kit (ODK) program installed on data collection tablets running on the Android operating system. Interviews were conducted in the respondent's preferred language in face-to-face computer-assisted personal interviewing (CAPI). The questionnaires were administered by extensively trained field teams who were fluent in these native languages and played a key role in validating the translations and contextualizing the questionnaires to improve the accuracy of data collected.

### Measures

**Experiences of VAC.** Experiences of VAC (sexual, physical, or emotional) were captured using questions posed to individuals aged 13–17 years on whether they had 'ever experienced' any of these forms of violence for lifetime exposure, and to those aged 18–24 years on their experiences of violence 'before the age of 18'.

*Sexual violence* included having experienced one or more incidents of unwanted sexual touching; attempted forced sex; pressured or coerced sex; and, physically forced sex, perpetrated by any person.

*Physical violence* included having experienced one or more incidents of slapping, pushing, shoving, shaking, or of having something thrown at them to intentionally hurt them; punching, kicking, whipping, or being beaten with an object; choking, smothering, a person trying to drown them, or burning them intentionally; and, using or threatening them with a knife, gun or other weapon, perpetrated by an intimate partner, peer, parent or adult caregiver or other adult relative, and/or other adults in the community.

*Emotional violence* included having experienced one or more incidents of being told that they were not loved or did not deserve to be loved; being told that they should never have been born or should have died; and, being ridiculed or put down, for example, being told that they were stupid or useless, perpetrated by a parent or adult caregiver or other adult relative, an intimate partner, or peer.

**Witnessing physical violence.** This was measured based on questions to participants about witnessing physical violence perpetrated in their shelter/home and often by a father/stepfather, or outside the family environment (in the community/neighbourhood) by people they knew well or strangers. Those aged 13–17 years were asked if they had witnessed the violence at any time in their life, while those aged 18–24 years they were asked if they had witnessed instances of violence before the age of 18 years. Possible responses were never, once, or more than one time. Those who had witnessed any physical violence once or more than once were categorized as having witnessed violence.

### Outcome variables

**Polyvictimization.** The current analysis examined lifetime victimisation across physical, sexual, and emotional violence, as well as witnessing violence in the home and community. Each violence outcome was coded as '1' if the respondent reported experiencing any of the items within each category, and '0' otherwise. Lifetime polyvictimization was operationalized as an additive index created by summing the disaggregated types of reported lifetime physical, sexual, emotional violence, and witnessing violence in the home and/or community. Participants who reported experiencing two or more types of victimization (out of five possible types) were categorized as polyvictims, while those who reported experiencing zero or one type of victimization were considered non-polyvictims. Studies using a similar approach have been published [9,10].

**Mental health.** Mental health was assessed based on three measures.

a) *Severe mental distress*, which was generated using six questions from the Kessler Psychological Distress Scale (K6) [30] that included how often the participant felt nervous; hopeless; restless; so sad that nothing could cheer them up; that everything was an effort; or worthless, all the time, most of the time, some of the time, a little of the time, or none of the time, during the past 30 days. Responses to each of these six questions for any participant were scored between 0 (for none of the time) and 4 (for all of the time) and summed for a total possible score of between 0 and 24 points. The resulting scores were then categorized into 'none or less severe mental distress' for scores of less than 13, and 'moderate to severe mental distress' for scores of 13 points or higher [31].

b) *Suicidal ideation and/or attempted suicide* was assessed using responses to questions about whether the participant had ever thought about killing themselves or ever tried to kill themselves. Respondents who answered 'yes' to either question were categorized as reporting suicidal ideation and/or as having attempted suicide.

c) *Self-harm* was generated from the question on whether the participant had ever intentionally hurt themselves in any way. Those who answered 'yes' to this question were categorized as reporting self-harm.

### Independent variables

a) **Individual-level factors;** Included i) age, categorized into two groups (1 = 13–17 years and 2 = 18–24 years); ii) school enrolment (coded as 0 = not enrolled and 1 = currently enrolled in school); iii) Disability status (coded as 1 = with

disability and 0 otherwise), Disability was assessed based on the Washington Group Short Set on Functioning (WG-SS) questions that determine whether individuals have difficulty performing basic activities such as seeing, hearing [which was excluded from the Uganda and Ethiopia HVACS], walking, cognition, self-care, and communication [32]. For the purposes of our analyses, the sub-population 'with disabilities' includes everyone with at least one domain that is coded as having 'some difficulty', 'a lot of difficulty', and/or 'or cannot do it at all'; and iv) Endorsement of intimate partner violence against women (IPVAW): We considered questions administered to 13–24-year-olds on whether they believed that a husband would be justified in hitting or beating his wife if she (a) went out without informing him; (b) neglected the children; (c) argued with him; (d) refused to have sex with him; and (e) burned the food. A response of 'yes' to any one of the five questions (a-e) was coded as 1 (endorsed IPVAW), otherwise, 'no'.

b) **Family-level factors:** We included orphanhood, described as having lost one both biological parents; household food insecurity, described as not thinking that their household had enough money for food; relationship with mother (or father), described as reporting that they either had a very close or easy relationship, or a difficult or very difficult relationship with their mother (or father); and living in a female-headed household.

c) **Community-level or extra-familial factors:** We included having supportive friendships, described as having reliable and supportive friends to share experiences, provide help, and discuss problems; group participation in the community, described as participation in clubs, sports teams, religious organizations, or other groups in their community; and supportive adult in the community, described as the presence of a trusted, caring adult outside the home and school who provided emotional support, encouragement, and motivation.

## Ethical considerations

The Uganda HVACS was approved by the Population Council Institutional Review Board (Protocol 986 dated 21 October 2021) and Mildmay Uganda Research Ethics Committee (MUREC), REF 0310–2021 dated 24 November 2021. The research was also granted regulatory approval by the Uganda National Council for Science and Technology (UNCST), REF SS1130ES dated 10 January 2022. The Ethiopia HVACS was approved by the Population Council Institutional Review Board (Protocol 986 dated 21 October 2021) and the Ethiopian Public Health Association (EPHA) Institutional Review Board, EPHA/OG/789/23 dated 10 August 2023. The Ethiopia Refugees and Returnees Service (RRS) also provided administrative authorisation to enter the camps. IRBs and Research Ethics Committees in both countries approved the entire research protocol, including the consent forms and the consenting process described below.

All participants provided verbal informed consent (recorded electronically) – for those aged 18–24 years and emancipated minors aged 13–17 years who had assumed adult roles and responsibilities including household headship, marriage, and/or procreation. Parental consent and assent were obtained for those aged 13–17 years. In such cases, and to protect any participants whose guardian(s) were the perpetrators of violence, their parents or caregivers were given limited information about the objectives of the research and rather told that it was focused on the health, educational, and life experiences of children and young people in refugee settings.

A trauma response plan for participants requiring such supports was included as part of the Uganda and Ethiopia HVACS implementation plan. It included offering counselling and voluntary referrals through proximal support agencies in place in these refugee settings. These supportive services were delivered by trained case workers who accompanied the team throughout the fieldwork period. Psychosocial support was extended to any member of the household requiring it.

## Analysis

The analyses utilize data from 1,338 females and 927 males aged 13–24 years from Uganda's refugee settlements and 1,937 females and 1,536 males aged 13–24 years from Ethiopia's refugee camps who completed the survey. We opted

not to pool and rather present country-specific analyses because this study aims to highlight contextual differences that are critical for programming and policy.

We used cross-tabulation with Pearson Chi-Square ($\chi^2$) to examine significant differences in the distribution of female and male respondents by a) individual factors (i.e., age, disability status, endorsement of IPVAW, and school enrolment); b) family factors (i.e., orphanhood status, household food insecurity, difficult relationship with mother or father, and living in a female-headed household); and 3) community factors (i.e., having a supportive friendships or adult in the community, and participating in community groups); and experiencing polyvictimization. We also estimated adjusted logistic regression models to examine the magnitude and direction of the association between these individual-, family-, and community-level factors and polyvictimization. Additionally, we used unadjusted (Model 1) and adjusted (Model 2) logistic regression models to examine the association between polyvictimization and mental health outcomes (moderate-to-severe mental distress, self-harm, and suicidal ideation/attempt). The analyses were stratified by sex and country and considered the complex survey design by applying weights to estimate prevalence among the sub-groups examined and adjusting standard errors for complex sampling. We considered a p-value of <0.05 statistically significant, with 95% confidence intervals. Correlation and Variance Inflation Factor (VIF) analyses were performed to check for multicollinearity and tolerance values for all variables. No major multicollinearity problems were associated with most variables (mean VIF range 1.12–1.16 for Uganda and 1.11–1.21 for Ethiopia). All analyses were conducted using Stata® version 18.

## Results

### Background characteristics

In Uganda, there were no major statistically significant differences in the background characteristics between females and males, except at the family and community levels (Table 1). A significantly higher proportion of females than males reported living in female-headed households (72% vs. 44%) and having a supportive adult in the community who provides emotional support, encouragement, and motivation (29% vs. 19%). More generally, a majority of participants were enrolled in school (65% of females, 73% of males) and reported household food insecurity (52% of females, 67% of males), and just over a quarter were orphans (26% of females, 30% of males). For context, national estimates in Uganda showed that 31% of households were headed by females in 2023/24, with the primary and secondary school gross enrolment rates (GERs) at 120% and 34% in 2023/24 [33].

In Ethiopia, significantly more males than females reported being enrolled in school (68% vs. 52%). There were no other notable differences in the background characteristics between females and males, except at the community level. Significantly more males than females reported having supportive friendships (50% vs. 41%), participating in community groups within camps (28% vs. 6%) and having a supportive adult in the community (50% vs. 30%). A high number of female (86%) and male (81%) respondents reported living in a female-headed households. For context, national estimates from Ethiopia indicate that 22% of household were female headed in 2019 [34], and that the primary and secondary GERs were 96.1% and 14.6% respectively in 2022/23 [35].

### Prevalence of victimization, polyvictimization, and mental health outcomes

In Uganda, significantly higher proportions of males compared to females had experienced physical violence (49% vs. 26%), emotional violence (25% vs. 18%), witnessed physical intimate partner violence against their mother (30% vs. 19%), witnessed physical violence in the community (41% vs. 19%), and had experienced two or more types of violence (49% vs. 30%) (Table 2).

In Ethiopia, a significantly higher percentage of females (10%) than males (3%) had experienced sexual violence, while a higher proportion of males (35%) compared to females (24%) had witnessed physical violence in the community. About a third of both females (29%) and males (33%) reported experiencing two or more types of violence.

**Table 1. Background characteristics of 13-24-year-old females and males, Uganda HVACS 2022 and Ethiopia HVACS 2024.**

| | Uganda | | | Ethiopia | | |
| --- | --- | --- | --- | --- | --- | --- |
| | Females (N = 1338)[a] | Males (N = 997)[a] | | Females (N = 1937)[a] | Males (N = 1536)[a] | |
| | Weighted%[95%CI] | Weighted%[95%CI] | p-value[b] | Weighted%[95%CI] | Weighted%[95%CI] | p-value[b] |
| **Individual Factors** | | | | | | |
| **Age** | | | | | | |
| 13-17 | 51.4[47.2,55.6] | 48.5[44.5,52.5] | 0.319 | 52.4[49.0,55.8] | 51.4[48.1,54.8] | 0.679 |
| 18-24 | 48.6[44.4,52.8] | 51.5[47.5,55.5] | | 47.6[44.2,51.0] | 48.6[45.2,51.9] | |
| **Disability status** | | | | | | |
| Without disability | 77.4[73.1,81.2] | 69.0[59.5,77.2] | 0.096 | 85.4[75.7,91.6] | 85.5[78.0,90.7] | 0.991 |
| With disability | 22.6[18.8,26.9] | 31.0[22.8,40.5] | | 14.6[8.4,24.3] | 14.5[9.3,22.0] | |
| **Endorsement of IPVAW** | | | | | | |
| No | 42.3[33.6,51.5] | 53.0[39.3,66.3] | 0.272 | 37.7[29.4,46.7] | 33.4[26.6,40.9] | 0.443 |
| Yes | 57.7[48.5,66.4] | 47.0[33.7,60.7] | | 62.3[53.3,70.6] | 66.6[59.1,73.4] | |
| **School enrolment** | | | | | | |
| No | 34.7[28.6,41.5] | 26.7[20.7,33.8] | 0.177 | 48.0[43.8,52.2] | 31.9[27.5,36.7] | **<0.001** |
| Yes | 65.3[58.5,71.4] | 73.3[66.2,79.3] | | 52.0[47.8,56.2] | 68.1[63.3,72.5] | |
| **Family Factors** | | | | | | |
| **Orphanhood by age 18** | | | | | | |
| No | 74.2[67.6,79.8] | 70.2[65.3,74.6] | 0.335 | 80.1[75.4,84.0] | 78.5[68.1,86.2] | 0.755 |
| Yes | 25.8[20.2,32.4] | 29.8[25.4,34.7] | | 19.9[16.0,24.6] | 21.5[13.8,31.9] | |
| **Household food insecurity** | | | | | | |
| No | 48.1[35.4,61.0] | 33.3[22.5,46.1] | | 16.7[11.9,22.9] | 23.6[19.3,28.7] | 0.069 |
| Yes | 51.9[39.0,64.6] | 66.7[53.9,77.5] | 0.201 | 83.3[77.1,88.1] | 76.4[71.3,80.7] | |
| **Relationship with mother** | | | | | | |
| Very close/easy | 87.4[80.2,92.2] | 86.4[79.7,91.2] | | 88.0[84.2,91.0] | 87.2[77.0,93.3] | 0.858 |
| Difficult/very difficult | 12.6[7.8,19.8] | 13.6[8.8,20.3] | 0.831 | 12.0[9.0,15.8] | 12.8[6.7,23.0] | |
| **Relationship with father** | | | | | | |
| Very close/easy | 74.9[67.4,81.2] | 79.7[70.8,86.5] | 0.417 | 80.9[75.5,85.3] | 79.2[68.4,87.0] | 0.739 |
| Difficult/very difficult | 25.1[18.8,32.6] | 30.3[13.5,29.2] | | 19.1[14.7,24.5] | 20.8[13.0,31.6] | |
| **Female-headed household** | | | | | | |
| No | 28.1[20.6,37.0] | 56.3[46.7,65.4] | **<0.001** | 14.2[8.9,22.0] | 18.9[13.6,25.6] | 0.298 |
| Yes | 71.9[63.0,79.4] | 43.7[34.6,53.3] | | 85.8[78.0,91.1] | 81.1[74.4,86.4] | |
| **Community Factors** | | | | | | |
| **Supportive friendships** | | | | | | |
| No | 52.9[46.3,59.3] | 59.5[51.2,67.4] | 0.272 | 59.3[52.9,65.3] | 48.6[42.9,54.3] | **0.016** |
| Yes | 47.1[40.7,53.7] | 40.5[32.6,48.8] | | 40.7[34.7,47.1] | 51.4[45.7,57.1] | |
| **Community group participation** | | | | | | |
| No | 90.1[86.3,92.9] | 86.9[80.2,91.5] | 0.376 | 94.2[90.6,96.5] | 72.1[66.0,77.5] | **<0.001** |
| Yes | 9.9[7.1,13.7] | 13.1[8.5,19.8] | | 5.8[3.5,9.4] | 27.9[22.5,34.0] | |
| **Supportive adult in the community** | | | | | | |
| No | 70.7[65.2,75.6] | 80.6[74.4,85.7] | **0.013** | 70.2[63.8,75.8] | 49.8[42.0,57.5] | **<0.001** |
| Yes | 29.3[24.4,34.8] | 19.4[14.3,25.6] | | 29.8[24.2,36.2] | 50.2[42.5,58.0] | |

**Note:** [a]-Unweighted denominators; [b]-Pearson Chi-square p-values.

**Table 2. Prevalence of victimization, polyvictimization, and mental health outcomes among 13-24-year-old females and males, Uganda HVACS 2022 and Ethiopia HVACS 2024.**

| | Uganda | | | Ethiopia | | |
|---|---|---|---|---|---|---|
| | Females (N = 1338)[a] | Males (N = 997)[a] | | Females (N = 1937)[a] | Males (N = 1536)[a] | |
| | Weighted%[95%CI] | Weighted%[95%CI] | p-value[b] | Weighted%[95%CI] | Weighted%[95%CI] | p-value[b] |
| **Lifetime violence outcomes** | | | | | | |
| Childhood physical violence | 26.3[22.9,30.0] | 48.6[39.9,57.3] | **<0.001** | 27.3[20.8,34.9] | 30.8[21.4,42.1] | 0.567 |
| Childhood sexual violence | 13.9[7.6,23.9] | 9.9[6.5,14.9] | 0.310 | 9.8[8.1,11.9] | 3.4[2.1,5.4] | **<0.001** |
| Childhood emotional | 18.1[13.8,23.4] | 25.3[20.3,31.1] | **0.042** | 16.8[12.8,21.8] | 20.6[13.0,31.1] | 0.436 |
| Witnessed physical intimate partner violence at home | 19.0[13.2,26.7] | 30.5[21.7,41.1] | **0.026** | 28.6[24.2,33.5] | 27.9[20.1,37.4] | 0.891 |
| Witnessed physical violence in the community | 19.0[14.0,25.4] | 40.6[31.6,50.4] | **0.002** | 23.8[19.4,28.9] | 35.3[27.7,43.6] | **0.014** |
| **Number of victimisation** | | | | | | |
| 0 | 49.2[40.5,58.0] | 32.4[24.4,41.4] | | 44.7[39.1,50.5] | 42.6[34.1,51.7] | |
| 1 | 21.0[18.2,24.0] | 18.2[15.2,21.8] | | 26.1[21.6,31.3] | 24.0[19.1,29.8] | |
| 2 | 19.6[16.0,23.8] | 25.2[21.4,29.4] | **0.004** | 14.7[11.3,18.8] | 16.0[12.0,21.1] | 0.478 |
| 3 or more | 10.3[6.7,15.4] | 24.2[17.9,31.9] | | 14.4[11.0,18.7] | 17.3[10.1,27.9] | |
| **Polyvictimization** | | | | | | |
| Otherwise | 70.2[63.2,76.3] | 50.6[41.3,59.8] | **0.005** | 70.9[64.7,76.4] | 66.7[55.0,76.6] | 0.486 |
| Polyvictims | 29.8[23.7,36.8] | 49.4[40.2,58.7] | | 29.1[23.6,35.3] | 33.3[23.4,45.0] | |
| **Mental health outcomes** | | | | | | |
| Moderate/severe mental distress | 69.4[64.0,74.3] | 57.4[45.1,68.7] | 0.074 | 66.3[60.1,72.0] | 55.3[41.7,68.1] | 0.129 |
| Suicide ideation/attempt | 7.0[4.4,10.9] | 6.4[4.0,10.1] | 0.763 | 8.5 [6.3,11.2] | 9.7[4.3,20.5] | 0.744 |
| Self-harm | 7.0[5.8,10.8] | 6.9[3.7,12.3] | 0.611 | 7.0[5.0,9.8] | 7.4[3.1,17.0] | 0.894 |

**Note:** [a]-Unweighted denominators; [b]-Pearson Chi-square p-values.

There were no significant differences in the prevalence of mental health outcomes between females and males in either setting, however, slightly more females (69% in Uganda, 66% in Ethiopia) than males (57% in Uganda, 55% in Ethiopia) reported experiencing moderate to severe mental distress in the past 30 days.

## Prevalence of polyvictimization by background characteristics

In Uganda, among both females and males, polyvictimization was more prevalent among those who endorsed IPVAW compared to those who did not (34% vs. 24% for females; 61% vs. 39% for males), and those who reported difficult relationships with their fathers compared to those in close relationships with their fathers (37% vs. 28% for females and 73% vs. 44% for males) (Table 3). In contrast to males, polyvictimization was more prevalent in females with disabilities compared to those without (44% vs. 26%), those who reported difficult relationships with their mothers compared to those in close relationships with their mothers (39% vs. 29%), and those with no group participation in their community compared to those who participated (31% vs. 16%), as well as among those who lacked a supportive adult in the community compared to those that did (33% vs. 23%).

In Ethiopia, among females, significantly higher percentage of polyvictimization was reported among those who were orphaned compared to those who were not (40% vs. 26%) and among those not enrolled in school compared to those that were currently enrolled (35% vs. 24%). Among males, a significantly higher percentage of polyvictimization among those with disabilities compared to those without (59% vs. 29%), among those who reported difficult relationships with their mother (72% vs. 29%) or father (57% vs. 27%) compared to those with close relationships. Additionally, a higher

**Table 3. Percentage of 13-24-year-old females and males who experienced polyvictimization by background characteristics, Uganda HVACS 2022 and Ethiopia HVACS 2024.**

| | Uganda | | | | | | Ethiopia | | | | | |
|---|---|---|---|---|---|---|---|---|---|---|---|---|
| | **Females** | | | **Males** | | | **Females** | | | **Males** | | |
| | N[a] | % Polyvictims | p-value | N[a] | % Polyvictims | p-value | N[a] | % Polyvictims | p-value | N[a] | % Polyvictims | p-value |
| **Age** | | | | | | | | | | | | |
| 13-17 | 716 | 27.0[20.1,35.2] | 0.109 | 532 | 52.7[41.9,63.2] | 0.107 | 996 | 28.3[21.8,36.0] | 0.583 | 854 | 36.2[24.9,49.3] | 0.177 |
| 18-24 | 622 | 38.2[25.5,41.1] | | 395 | 46.3[37.1,55.8] | | 941 | 30.0[24.1,36.6] | | 682 | 30.3[19.9,43.1] | |
| **Disability Status** | | | | | | | | | | | | |
| Without disability | 1027 | 25.7[20.3,31.9] | **<0.001** | 671 | 47.4[33.4,61.7] | 0.567 | 1745 | 29.0[23.6,35.2] | 0.960 | 1289 | 28.9[18.7,41.9] | **<0.001** |
| With disability | 311 | 44.1[32.3,56.7] | | 256 | 54.0[40.3,67.1] | | 192 | 29.5[14.3,51.3] | | 247 | 59.4[47.2,70.5] | |
| **Endorsement of IPVAW** | | | | | | | | | | | | |
| No | 696 | 24.3[18.6,31.0] | **0.005** | 529 | 39.0[28.0,51.3] | **0.008** | 763 | 24.1[16.1,34.3] | 0.105 | 530 | 23.2[09.4,46.8] | 0.207 |
| Endorsed | 642 | 33.9[25.9,43.0] | | 398 | 61.1[51.9,69.7] | | 1174 | 32.2[26.4,38.5] | | 1006 | 33.3[23.1,45.4] | |
| **School enrolment** | | | | | | | | | | | | |
| Not enrolled | 534 | 29.9[19.5,42.9] | 0.994 | 276 | 44.4[29.2,60.8] | 0.442 | 914 | 34.8[27.9,42.5] | **0.040** | 475 | 28.9[19.0,41.4] | 0.197 |
| Currently enrolled | 804 | 29.8[23.4,37.1] | | 651 | 51.2[40.8,61.5] | | 1023 | 23.8[16.5,33.0] | | 1061 | 35.4[24.0,48.8] | |
| **Orphanhood by 18** | | | | | | | | | | | | |
| No | 978 | 28.7[21.8,36.8] | 0.182 | 648 | 50.7[40.0,61.3] | 0.516 | 1550 | 26.4[20.7,33.0] | **0.002** | 1298 | 31.5[20.9,44.4] | 0.272 |
| Yes | 360 | 33.1[26.4,40.6] | | 279 | 46.3[34.7,58.4] | | 387 | 40.1[31.4,49.5] | | 238 | 40.2[25.8,56.6] | |
| **Household food insecurity** | | | | | | | | | | | | |
| No | 548 | 25.5[19.0,33.3] | 0.072 | 340 | 54.3[48.5,59.9] | 0.236 | 292 | 25.3[18.5,33.5] | 0.305 | 350 | 35.2[22.4,50.6] | 0.648 |
| Yes | 690 | 33.9[26.2,42.5] | | 587 | 47.0[34.8,59.6] | | 1645 | 29.9[23.6,37.0] | | 1186 | 32.8[22.5,45.0] | |
| **Relationship with mother** | | | | | | | | | | | | |
| Very close/easy | 1195 | 28.5[22.5,35.3] | **0.027** | 837 | 47.2[36.5,58.2] | 0.233 | 1738 | 28.4[22.9,34.7] | 0.191 | 1387 | 27.8[18.3,39.8] | **<0.001** |
| Difficult/very difficult | 143 | 39.2[28.0,51.7] | | 90 | 63.5[39.7,82.1] | | 199 | 34.2[23.8,46.3] | | 149 | 71.2[53.4,84.3] | |
| **Relationship with father** | | | | | | | | | | | | |
| Very close/easy | 1025 | 27.5[20.3,36.1] | **0.036** | 758 | 43.5[34.2,53.3] | **0.015** | 1612 | 27.5[22.6,33.0] | 0.060 | 1285 | 27.1[18.4,37.9] | **0.001** |
| Difficult/very difficult | 313 | 36.8[29.9,44.2] | | 169 | 72.7[48.0,88.5] | | 325 | 35.8[24.2,49.4] | | 251 | 57.3[38.6,74.0] | |
| **Female-headed household** | | | | | | | | | | | | |
| No | 437 | 34.6[24.5,46.2] | 0.138 | 490 | 50.3[42.4,58.2] | 0.757 | 348 | 36.6[26.9,47.5] | 0.087 | 427 | 44.7[29.3,61.1] | **0.004** |
| Yes | 900 | 28.0[22.2,34.7] | | 437 | 48.3[34.2,62.7] | | 1572 | 27.9[22.1,34.5] | | 1084 | 31.1[21.9,42.1] | |
| **Supportive friendships** | | | | | | | | | | | | |
| No | 721 | 31.3[22.5,41.6] | 0.573 | 590 | 48.5[39.6,57.6] | 0.732 | 1197 | 28.2[21.7,35.9] | 0.607 | 863 | 35.9[25.6,47.6] | 0.102 |
| Yes | 617 | 28.2[21.3,36.4] | | 337 | 50.7[37.0,64.3] | | 740 | 30.4[23.3,38.6] | | 673 | 31.0[20.3,44.1] | |
| **Community group participation** | | | | | | | | | | | | |
| No | 1220 | 31.3[24.7,38.9] | **0.003** | 817 | 49.7[39.6,59.7] | 0.786 | 1824 | 29.3[23.4,36.0] | 0.821 | 1190 | 30.2[21.2,41.1] | 0.109 |

*(Continued)*

**Table 3.** (Continued)

| | Uganda | | | | | | Ethiopia | | | | | |
|---|---|---|---|---|---|---|---|---|---|---|---|---|
| | Females | | | Males | | | Females | | | Males | | |
| | Nª | % Polyvictims | p-value | Nª | % Polyvictims | p-value | Nª | % Polyvictims | p-value | Nª | % Polyvictims | p-value |
| Yes | 118 | 16.3[10.6,24.2] | | 110 | 47.9[36.8,59.1] | | 113 | 26.2[08.3,58.2] | | 346 | 41.4[24.2,61.0] | |
| **Supportive adult in the community** | | | | | | | | | | | | |
| No | 922 | 32.7[25.2,41.2] | **0.020** | 712 | 50.0[41.2,59.8] | 0.206 | 1348 | 28.4[21.3,36.7] | 0.665 | 877 | 37.9[25.8,51.8] | **0.003** |
| Yes | 416 | 23.0[17.4,29.7] | | 215 | 44.8[34.1,56.1] | | 589 | 30.8[22.9,40.0] | | 659 | 28.8[19.8,39.8] | |

**Note:** ª-Unweighted denominators.

percentage of polyvictimization was reported among males in non-female-headed households compared to those in female-headed households (45% vs. 31%) and those reporting having no supportive adult in the community compared to those who had such supportive relationships (38% vs. 29%).

## Factors associated with polyvictimization

In Uganda, individual and family factors were associated with reporting polyvictimization. Both females and males had higher odds of experiencing polyvictimization if they had a disability (AOR = 1.8 for females and 1.5 for males), endorsed IPVAW (AOR = 1.6 for females and 2.6 for males), or had a difficult relationship with their father (AOR = 1.4 for females and 1.9 for males) compared to their counterparts (Table 4). Among females, living in a food-insecure household increased the odds of polyvictimization (AOR = 1.4), while living in a female-headed household significantly reduced the odds of polyvictimization by 30%, a trend not observed among males.

In Ethiopia, individual, family and community factors were significantly associated with reporting polyvictimization. Both females and males had higher odds of experiencing polyvictimization if they had a disability (AOR = 1.5 for females and 2.8 for males), endorsed IPVAW (AOR = 2.0 for females and 1.5 for males), or had a difficult relationship with their father (AOR = 1.4 for females and 1.7 for males) compared to their counterparts (Table 4). Conversely, both females and males had lower odds of polyvictimization if they were aged 18–24 (AOR = 0.6 for females and males) compared to those aged 13–17 years, or if they lived in a female-headed household (AOR = 0.7 for females and 0.5 for males). Among females, being in school and participating in community groups reduced the odds of polyvictimization by 40% and 50%, respectively. Among males, living in a food-insecure household and having a difficult relationship with mother increased the odds of polyvictimization by 1.4 and 2.7 times, respectively.

As shown in Table 5, among both females and males in Uganda, polyvictimization was associated with higher odds of mental distress (AOR = 3.3 for females and 1.5 for males), self-harm (AOR = 1.9 for females and 2.1 for males) and suicide ideation/attempts (AOR = 3.0 for females and 4.6 for males) compared to non-polyvictims. Similarly in Ethiopia, polyvictimized females and males had higher odds of mental depression (AOR = 2.4 for females and 1.7 for males), self-harm (AOR = 7.4 for females and 3.7 for males) and suicide ideation/attempts (AOR = 8.4 for females and 3.9 for males) compared to non-polyvictims.

## Discussion

This paper examined the prevalence of childhood polyvictimization, assessing the individual, family, and community-level risk and protective factors associated with it, as well as the mental health outcomes linked to polyvictimization among young people aged 13–24 years living in refugee settings in Uganda and Ethiopia.

**Table 4. Multivariate logistic regression examining factors associated with polyvictimization among 13-24-years-old females and males, Uganda HVACS 2022 and Ethiopia HVACS 2024.**

| | Uganda | | | | Ethiopia | | | |
|---|---|---|---|---|---|---|---|---|
| | Females | | Males | | Females | | Males | |
| Outcome: Polyvictimization | AOR[95% CI] | p-value | AOR[95% CI] | p-value | AOR[95% CI] | p-value | AOR[95% CI] | p-value |
| Aged 18–24 years | 1.03[0.76,1.40] | 0.825 | 1.16[0.84,1.60] | 0.374 | 0.63[0.50,0.79] | **0.000** | 0.61[0.48,0.78] | **0.000** |
| Has a disability | 1.82[1.38,2.40] | **0.000** | 1.48[1.09,2.02] | **0.013** | 1.53[1.11,2.12] | **0.010** | 2.81[2.06,3.84] | **0.000** |
| Endorsed norms justifying IPVAW | 1.60[1.25,2.05] | **0.000** | 2.61[1.97,3.45] | **0.000** | 1.95[1.56,2.44] | **0.000** | 1.48[1.15,1.90] | **0.002** |
| Currently enrolled in school | 0.89[0.65,1.23] | 0.482 | 1.17[0.84,1.65] | 0.354 | 0.62[0.49,0.77] | **0.000** | 1.19[0.91,1.55] | 0.214 |
| Orphaned | 1.10[0.83,1.46] | 0.507 | 0.76[0.56,1.04] | 0.083 | 1.43[1.11,1.83] | **0.005** | 0.88[0.63,1.22] | 0.433 |
| Household food insecurity | 1.43[1.12,1.85] | **0.005** | 0.77[0.58,1.02] | 0.072 | 1.11[0.82,1.50] | 0.494 | 1.39[1.03,1.88] | **0.030** |
| Difficult relationship with mother | 1.32[0.88,1.97] | 0.173 | 1.13[0.69,1.85] | 0.614 | 1.27[0.90,1.79] | 0.166 | 2.72[1.81,4.07] | **0.000** |
| Difficult relationship with father | 1.40[1.04,1.89] | **0.028** | 1.90[1.31,2.76] | **0.001** | 1.35[1.02,1.79] | **0.035** | 1.71[1.24,2.35] | **0.001** |
| Female-headed household | 0.71[0.55,0.93] | **0.012** | 0.97[0.73,1.28] | 0.807 | 0.65[0.50,0.84] | **0.001** | 0.50[0.39,0.64] | **0.000** |
| Has supportive friends | 0.98[0.90,1.06] | 0.586 | 1.22[0.89,1.66] | 0.213 | 1.20[0.96,1.50] | 0.115 | 0.84[0.62,1.13] | 0.249 |
| Membership to community groups | 1.00[0.89,1.12] | 0.994 | 1.22[0.79,1.89] | 0.363 | 0.46[0.27,0.78] | **0.004** | 1.18[0.86,1.61] | 0.296 |
| Has a trusted adult in the community | 1.02[0.94,1.09] | 0.699 | 1.09[0.77,1.54] | 0.615 | 0.90[0.71,1.15] | 0.408 | 0.74[0.55,1.00] | 0.053 |

**Table 5. Bivariate and multivariate logistic regression examining the association between polyvictimization and poor mental health outcomes among 13-24-years-old females and males, Uganda HVACS 2022 and Ethiopia HVACS 2024.**

| | | Moderate/Severe mental distress | | Self-harm | | Suicide ideation/attempt | |
|---|---|---|---|---|---|---|---|
| | Polyvictimization | Model 1[a] | Model 2[b] | Model 1[a] | Model 2[b] | Model 1[a] | Model 2[b] |
| **Uganda** | | | | | | | |
| | None (ref) | | | | | | |
| Females | Polyvictims | 3.34[2.38,4.68]*** | 3.27[2.29,4.66]*** | 2.25[1.52,3.32]*** | 1.92[1.27,2.90** | 3.04[2.02,4.57]*** | 3.01[1.94,4.68]*** |
| | None (ref) | | | | | | |
| Males | Polyvictims | 1.31[1.00,1.73]*** | 1.51[1.12,2.04]** | 2.15[1.22,3.79]** | 2.13[1.17,3.87]* | 4.16[2.11,8.22]*** | 4.58[2.22,9.48]*** |
| **Ethiopia** | | | | | | | |
| | None (ref) | | | | | | |
| Females | Polyvictims | 2.44[1.95,3.04]*** | 2.39[1.89,3.03]*** | 7.94[5.37,11.75]*** | 7.45[4.92,11.27]*** | 9.26[6.17,13.90]*** | 8.45[5.51,12.95]*** |
| | None (ref) | | | | | | |
| Males | Polyvictims | 1.44[1.16,1.79]*** | 1.69[1.33,2.15]*** | 6.23[4.05,9.56]*** | 3.68[2.24,6.02]*** | 9.70[5.36,17.56]*** | 3.90[1.95,7.81]*** |

**Note:** ***p < 0.001; **p < 0.01; *p < 0.05; [a]-**Model 1** is unadjusted; [b]-Model 2 includes individual-level factors (age, disability status, endorsement of IPVAW, school enrolment), family-level factors (orphanhood, living in a female-headed household, household food insecurity, relationship with mother/father), and community-level factors (supportive friendships, community group participation, supportive adult in the community).

Significant differences between females and males were observed in the prevalence of violence, including polyvictimization. In both settings, males generally reported higher levels of physical and emotional violence, as well as witnessing physical violence in the community. In contrast, females reported higher levels of sexual violence. Notably, a greater proportion of males (49%) compared to females (30%) in Uganda had experienced polyvictimization, while the rates in Ethiopia were more comparable, with 33% of males and 29% of females reporting polyvictimization. These findings align with findings in non-humanitarian settings, where some studies have reported significantly higher levels of polyvictimization among boys compared to girls [7,8]. However, in other contexts, such as among adolescents in Ethiopia, the prevalence of polyvictimization was similar for both girls (50%) and boys (53%) [9]. Taken together, in both countries, the high rates of polyvictimization (above 29%) are concerning, yet comparable to the overall estimate of 38.1% observed in LMICs (Le et al., 2018).

Largely, we found no significant gender differences in background characteristics across both contexts. A notable proportion of both males and females in refugee settings in both countries reported endorsing IPVAW, with rates ranging from 47% to 67%. This finding points to concerning levels of acceptance of IPVAW, indicating the need for focused efforts to address gender norms within these contexts, beginning early in the life course. Additionally, while less surprising given the challenging economic circumstances, a high percentage of participants reported experiencing household food insecurity, ranging from 52% to 83%, with higher levels observed in Ethiopia. High proportions of both male and female respondents also indicated living in female-headed households (ranging from 44% to 86%) and reported limited community group participation (ranging from 72% to 94%), possibly suggestive of absent or weakened familial and community ties in these settings.

Our findings offer nuanced insights into the factors associated with polyvictimization within these settings, highlighting both similarities and differences between the refugee contexts in Uganda and Ethiopia. Across both countries, individual-level factors, such as disability status and endorsement of IPVAW were significantly associated with childhood polyvictimization among females and males in both countries. Similar findings have been published, including a study in high income countries indicating that children with disabilities face higher risks of polyvictimization, with variations based on age and type of disability [36]. Children with disability, particularly those living in humanitarian settings, experience an increased risk of different forms of violence compared to those without disabilities [37]. School enrolment was protective against polyvictimization for females in Ethiopia. This finding is consistent with studies conducted in non-humanitarian settings. For example, surveys in LMICs have shown that school enrolment was a protective factor against lifetime polyvictimization for females in Haiti and males in Kenya, Haiti, and Cambodia [10].

At the family level, a difficult relationship with one's father was a risk factor that heightened the odds of childhood polyvictimization for both females and males in refugee settings in both countries, while a difficult relationship with one's mother was a risk factor for polyvictimization among males in refugee camps in Ethiopia. On the contrary, residing in a female-headed household acted as a protective factor, reducing the odds of polyvictimization for females in both countries and males in Ethiopia. These findings are consistent with studies conducted in non-humanitarian settings. For example, surveys in LMICs have shown that not living with one's biological mother was a risk factor for females in Kenya [10]. Household food insecurity was a risk factor for polyvictimization among females in Uganda and males in Ethiopia, and orphanhood increased the likelihood of polyvictimization among females in Ethiopia. However, at the community level, participating in community groups was protective against polyvictimization for females in Ethiopia.

Finally, after adjusting for individual-, family-, and community-level factors, we found that polyvictimization was significantly associated with higher odds of experiencing moderate to severe mental distress in the past 30 days, self-harm, and suicide ideation/attempts among 13–24-years-old females and males in both Uganda and Ethiopia refuge settings. The link between polyvictimization and severe mental health consequences has been well-documented in various settings [3,5,9,14,15,17]. Our findings further reinforce the hypothesis that polyvictimization is widespread and associated with increased risk of adverse health outcomes within refugee contexts, including poor mental health.

### Limitations

Our study has some limitations. Firstly, the cross-sectional nature of the HVACS datasets prevents us from establishing causal relationships for the observed associations. Secondly, to operationalize measures of individual victimization and polyvictimization, we focused on lifetime experiences without accounting for multiple occurrences, thereby overlooking the impact of repetitive victimization. Additionally, in defining polyvictimization, we did not assign weights to certain types or combinations of victimization that may be more harmful and traumatizing than others—such as physical forced sex compared to peer and sibling victimization —despite evidence indicating these differences [4]. Lastly, the relationship between polyvictimization and mental health outcomes may be influenced by other intra-individual factors (e.g., personality traits or

coping strategies) and external factors (e.g., non-victimization adversities or social support), which were not considered in this analysis.

Despite these limitations, our analysis is based on HVACS, an adaptation of well-regarded VAC survey, which has been implemented in multiple countries [38]. It sheds light on the prevalence of childhood polyvictimization, associated risk and protective factors in refugee settings, and contributes to a broader understanding of overlapping forms of violence faced by children in similar contexts. These findings are particularly relevant for professionals working with children and families, offering valuable insights to inform inter-sectoral approaches and fostering new, synergistic strategies for the more effective prevention and response to violence against children.

## Conclusion and implications for programming

In summary, our findings confirm that polyvictimization is prevalent among refugee populations in Uganda and Ethiopia, with a higher prevalence observed among males, particularly in Uganda. The high rates of polyvictimization, coupled with troubling levels of acceptance of IPVAW, challenging economic conditions, and weakened or absent familial and community ties, may exacerbate polyvictimization and its consequences, including increasing the difficulty in addressing its root causes. It is therefore crucial that efforts to tackle violence in refugee settings prioritise preventive measures alongside response strategies. A comprehensive approach is needed – one that addresses individual, familial, and community factors, tackles gender norms, promotes economic empowerment for children and youth, and fosters stronger familial and community connections – starting earlier in the life course.

## Acknowledgments

We are sincerely grateful for the invaluable technical and logistical support provided by several organizations and partners throughout the design and implementation of this study. We acknowledge the U.S. Centers for Disease Control and Prevention (CDC), the Together for Girls Partnership, the Office of the Prime Minister – Department of Refugees in Uganda, the Refugees and Returnees Service in Ethiopia, the UNHCR Regional Bureau for East and Horn of Africa and the Great Lakes Region, UNHCR Uganda, UNHCR Ethiopia, and the dedicated UNHCR implementing partners. We are especially thankful to the refugee and host community members who generously contributed their time and perspectives, both as data collectors and as respondents. Their courage and collaboration have been integral to the success of this study.

## Author contributions

**Conceptualization:** Stella Muthuri, George Odwe, Francis Obare, Peter Kisaakye, Yohannes Wado, Caroline Kabiru, Chi-Chi Undie.

**Data curation:** Stella Muthuri, George Odwe, Francis Obare, Peter Kisaakye.

**Formal analysis:** Stella Muthuri, George Odwe, Francis Obare, Peter Kisaakye.

**Funding acquisition:** Francis Obare, Caroline Kabiru, Chi-Chi Undie.

**Investigation:** Stella Muthuri, George Odwe, Francis Obare, Dagim Habteyesus, Gloria Seruwagi, Yohannes Wado, Yadeta Bacha, Bonnie Wandera, Caroline Kabiru, Chi-Chi Undie.

**Methodology:** Stella Muthuri, George Odwe, Francis Obare, Peter Kisaakye, Dagim Habteyesus, Gloria Seruwagi, Yohannes Wado, Yadeta Bacha, Bonnie Wandera, Caroline Kabiru, Chi-Chi Undie.

**Project administration:** Stella Muthuri, George Odwe, Francis Obare, Peter Kisaakye, Dagim Habteyesus, Gloria Seruwagi, Yohannes Wado, Yadeta Bacha, Bonnie Wandera, Caroline Kabiru, Chi-Chi Undie.

**Supervision:** Stella Muthuri, George Odwe, Francis Obare, Peter Kisaakye, Dagim Habteyesus, Gloria Seruwagi, Yohannes Wado, Yadeta Bacha, Bonnie Wandera, Caroline Kabiru, Chi-Chi Undie.

**Validation:** Stella Muthuri, George Odwe, Francis Obare, Gloria Seruwagi, Yohannes Wado, Yadeta Bacha, Bonnie Wandera, Caroline Kabiru, Chi-Chi Undie.

**Writing – original draft:** Stella Muthuri, George Odwe.

**Writing – review & editing:** Stella Muthuri, George Odwe, Francis Obare, Peter Kisaakye, Dagim Habteyesus, Gloria Seruwagi, Yohannes Wado, Yadeta Bacha, Bonnie Wandera, Caroline Kabiru, Chi-Chi Undie.

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
