## [Decision Letter · Decision Letter 0]

6 Nov 2025

Dear Dr. Muthuri,

Thank you for submitting your manuscript to PLOS ONE. After careful consideration, we feel that it has merit but does not fully meet PLOS ONE’s publication criteria as it currently stands. Therefore, we invite you to submit a revised version of the manuscript that addresses the points raised during the review process.

Having reviewed both the manuscript and the referee reports, I largely concur with the point raised by referees 1 and 3 that the findings for the adult population are hard to interpret and should probably be excluded.  First, we do not know if these individuals were displaced as chidlren.  Second (and more importantly), the various risk and protective factors are measured contemporaneously (in adulthood), not retrospectively (at the time of the abuse), and thus the findings cannot be interpreted as evidence that any particular characteristic had a protective effect vis-a-vis polyvictimization: the victimization could have occurred long before any particular characteristic was determined.  I suggest you drop these findings.  I also concur with multiple points raised by the referees noting that a more robust defense of the definition of polyvictimization is required; and I also find that the differences between males and females seem somewhat overemphasized in the text (or at least, are overemphasized without any clear conceptual framework for understanding these differences).  Please respond carefully to all of the comments provided by all three referees.

We look forward to receiving your revised manuscript.

Kind regards,

Jessica Leight, PhD

Academic Editor

PLOS ONE

(2) Please describe in your methods section how capacity to provide consent was determined for the participants in this study. Please also state whether your ethics committee or IRB approved this consent procedure. If you did not assess capacity to consent please briefly outline why this was not necessary in this case.

“This study was funded by the United Kingdom (UK) government through the Baobab Research Program Consortium (RPC).”

“This study was funded by the United Kingdom (UK) government through the Baobab Research Program Consortium (RPC), with the Population Council Inc as the primary awardee under contract number PO8612.

The funder did not play any role in the

design of the study, collection, analysis and interpretation of the data, writing of the manuscript, or the decision to submit the manuscript for publication.”

Reviewers' comments:

Reviewer's Responses to Questions

**Comments to the Author**

1. Is the manuscript technically sound, and do the data support the conclusions?

Reviewer #1: Yes

Reviewer #2: Yes

Reviewer #3: Partly

2. Has the statistical analysis been performed appropriately and rigorously?

Reviewer #1: No

Reviewer #2: Yes

Reviewer #3: No

3. Have the authors made all data underlying the findings in their manuscript fully available?

Reviewer #1: No

Reviewer #2: Yes

Reviewer #3: Yes

4. Is the manuscript presented in an intelligible fashion and written in standard English?

Reviewer #1: Yes

Reviewer #2: Yes

Reviewer #3: Yes

Reviewer #1: The paper is well-written and focuses on interesting issues that are prevalent in conflict-affected countries. My specific comments based on my review of the paper are provided below:

(i) The paper indicates a number of studies that focus on the prevalence of polyvictimisation in developing countries. The contributions of the paper to the existing literature could have strengthened to go beyond – undertaking another study in that focuses on humanitarian settings (Uganda and Ethiopia). The justifications provided for another study isn’t sufficient to warrant publishing the paper in a reputable journal like PLOS ONE.

(ii) In the discussion of the data sources, there are some repetitions in the discussions of the data between line 138-145, and 161-164.

(iii) In addition to using an additive index for polyvictimisation, the study could conside the use of Principal Component Analysis (PCA) or Multiple Correspondence Analysis (MCA) to generate an index for polyvictimisation.

(iv) In the description of mental health measure, the study generates a threshold for less severe mental health score if the mental health score is less than 13, and moderate to severe mental health stress for scores of 13 points or higher. How did the study arrive at this threshold? Please provide evidence to lend credence to the cut-off adopted.

(v) In the empirical analysis, in addition to using chi-square (X2), the paper could have used test of mean difference to obtain difference in outcomes between polyvictimisation, and non-polyvictimisation.

(vi) It would have been nice to present the econometric model used in the empirical analysis of the multivariate regression model. The econometric model should be presented such that the outcome (dependent variable), and the regressors are indicated.

(vii) The empirical analysis could have gone beyond multivariate logistic regression to include a further empirical analysis such as propensity score matching (PSM) method that use matching of observable characteristics to obtain robust results compared to multivariate logistic regression.

Reviewer #2: Review of the paper titled

“Prevalence of childhood polyvictimization, mental health outcomes, and associated risk and protective factors in Ethiopia and Uganda refugee settings”

by Muthuri et al., 2025

Question and Motivation

The paper seeks to understand whether polyvictimization is prevalent among children in humanitarian settings and to identify its associated risk and protective factors. This is a highly relevant topic, given that children in refugee contexts face multiple vulnerabilities that may differ substantially from those in non-humanitarian environments.

Contribution to the Literature

Polyvictimization during childhood has been linked to adverse mental-health and social outcomes later in life. While several studies have examined this issue in low- and middle-income countries, there is limited evidence from humanitarian contexts, where mechanisms and exposure dynamics may differ.

By focusing on refugee populations in Uganda and Ethiopia, this paper contributes new empirical evidence and provides useful insights into how risk and protective factors operate in displacement settings.

General Assessment

I found the paper well-written, well-motivated, and statistically rigorous. The analyses are transparent, and the presentation is clear and consistent. I recommend the paper for publication after the authors address the following key points of clarification and discussion.

Major Comments

1. Clarify the relevance of the refugee context

The paper would benefit from a stronger articulation of why the refugee context is analytically important. What mechanisms or structural conditions make victimization dynamics different from those in host communities? Without this framing, readers might assume that children in refugee camps face similar risks as other disadvantaged children.

- Where does the victimization primarily occur—within households, among peers, or through interactions with host-community members?

- If feasible, provide basic descriptive contrasts between the refugee sample and national statistics (e.g., share of female-headed households, poverty rates, school enrollment). Such comparisons would help contextualize the findings.

2. Clarify timing and setting of victimization

It is not entirely clear whether the episodes of polyvictimization occurred within the camps or prior to displacement. Because polyvictimization is defined as exposure before age 18, respondents aged 18–24 may have spent part of their childhood outside the camps.

- How long have these individuals been displaced?

- Could the higher prevalence among older respondents in Ethiopia reflect pre-displacement experiences rather than conditions in the camps?

This clarification is crucial for interpreting the age pattern in the results.

3. Disaggregate the forms of victimization and explore gender-norm mechanisms

Polyvictimization is defined as exposure to two or more types of victimization. The paper could benefit from disaggregating these experiences to show which combinations most commonly overlap (e.g., emotional–physical, physical–sexual). Such descriptive insights would deepen understanding of mechanisms behind the associations.

Moreover, the study surfaces a potentially important link between gender-norm attitudes (endorsement of IPVAW) and experiences of childhood victimization. The observed association between endorsing intimate-partner violence and being a polyvictim is intriguing yet counterintuitive. One might expect that individuals who normalize violence would under-report such experiences, yet the opposite pattern emerges. This tension might stem from the types of violence reported—certain forms (e.g., emotional or community-level) may be interpreted differently depending on gender-norm beliefs.

To probe this, the authors could:

- Produce a cross-tabulation between types of violence and respondent characteristics (gender, IPVAW endorsement, disability, household structure).

- Or, more ambitiously, replicate the analysis by defining polyvictimization as exposure to specific pairs of violence types (2 × 2)—for example, emotional + physical, physical + sexual, etc.

Either approach would enrich the discussion and clarify whether the IPVAW relationship reflects differences in perception, reporting, or actual exposure. I would be very interested in seeing these results.

4- Understanding the relationship between mental health and polyvictimization

The paper could more explicitly unpack what drives the association between mental health outcomes (e.g., self-harm, suicidal ideation or attempts) and polyvictimization. In Ethiopia, the point estimates drop by almost half once models are adjusted for gender, suggesting that compositional or mediating factors may be at play.

I encourage the authors to consider a sequential adjustment approach—adding individual-level factors first, then family-level, and finally community-level variables—to assess which dimension accounts for the observed attenuation. This would provide deeper insight into which factors most strongly mediate or confound the link between victimization and mental health and would strengthen the causal interpretation of the associations.

Minor Suggestions

- When reporting results, use neutral language such as “associated with higher/lower odds” instead of causal terms (“increased risk,” “reduced risk”).

- Visualization: The results section is dense with tables. Adding graphical summaries (e.g., bar charts or coefficient plots) would help readers grasp key patterns quickly. For instance, Table 2 contains more information than is discussed in the text; parts of it could be moved to an appendix while the main text features a concise visualization.

Reviewer #3: The exposition is generally good. It could be tightened in some places and expanded in others. The motivation and framing of the research question is area that requires attention. Some polishing and proof-reading could be beneficial, but is secondary. Please see attached report for more detailed comments.

**Do you want your identity to be public for this peer review?** For information about this choice, including consent withdrawal, please see our Privacy Policy

Reviewer #1: **Yes:** Dr Joseph Ajefu

Reviewer #2: No

Reviewer #3: No

---

## [Author Response · Author response to Decision Letter 1]

6 Jan 2026

Response to Reviewers - PONE-D-25-42079 file attached.

---

## [Decision Letter · Decision Letter 1]

5 Feb 2026

Prevalence of childhood polyvictimization, mental health outcomes, and associated risk and protective factors in Ethiopia and Uganda refugee settings.

PONE-D-25-42079R1

Dear Dr. Muthuri,

Thank you for your thorough response to the request for revisions.  We’re pleased to inform you that your manuscript has been judged scientifically suitable for publication and will be formally accepted for publication once it meets all outstanding technical requirements.

Kind regards,

Jessica Leight, PhD

Academic Editor

PLOS One

Additional Editor Comments (optional):

Reviewers' comments:

Reviewer's Responses to Questions

**Comments to the Author**

Reviewer #2: (No Response)

Reviewer #3: (No Response)

2. Is the manuscript technically sound, and do the data support the conclusions?

Reviewer #2: Partly

Reviewer #3: No

3. Has the statistical analysis been performed appropriately and rigorously?

Reviewer #2: Yes

Reviewer #3: No

4. Have the authors made all data underlying the findings in their manuscript fully available?

Reviewer #2: Yes

Reviewer #3: (No Response)

5. Is the manuscript presented in an intelligible fashion and written in standard English?

Reviewer #2: No

Reviewer #3: (No Response)

Reviewer #2: This revised version of the manuscript represents a clear improvement over the previous submission. The authors have addressed most of the comments raised in the first round in a careful and thoughtful manner.

Please find below my remaining comments regarding one issue that I believe would benefit from further consideration.

Reviewer #3: Unfortunately, the authors have not addressed the thoughtful comments raised by the editor and the reviewers. As a result, the results of the study are not reliable. Given the importance of the topic, the results must be extremely robust even for a correlation.

**Do you want your identity to be public for this peer review?** For information about this choice, including consent withdrawal, please see our Privacy Policy

Reviewer #2: No

Reviewer #3: No

---

## [Editor Report · Acceptance letter]

PONE-D-25-42079R1

PLOS One

Dear Dr. Muthuri,

I'm pleased to inform you that your manuscript has been deemed suitable for publication in PLOS One. Congratulations! Your manuscript is now being handed over to our production team.

Kind regards,

on behalf of

Dr. Jessica Leight

Academic Editor

PLOS One